# 4D Unsupervised Object Discovery

**Yuqi Wang**[1,2]    **Yuntao Chen**[3]    **Zhaoxiang Zhang**[1,2,3]

[1] Center for Research on Intelligent Perception and Computing (CRIPAC),
National Laboratory of Pattern Recognition (NLPR),
Institute of Automation, Chinese Academy of Sciences (CASIA)
[2] School of Artificial Intelligence, University of Chinese Academy of Sciences
[3] Centre for Artificial Intelligence and Robotics, HKISI_CAS

{wangyuqi2020,zhaoxiang.zhang}@ia.ac.cn
chenyuntao08@gmail.com

## Abstract

Object discovery is a core task in computer vision. While fast progresses have been made in supervised object detection, its unsupervised counterpart remains largely unexplored. With the growth of data volume, the expensive cost of annotations is the major limitation hindering further study. Therefore, discovering objects without annotations has great significance. However, this task seems impractical on still-image or point cloud alone due to the lack of discriminative information. Previous studies underlook the crucial *temporal* information and constraints naturally behind multi-modal inputs. In this paper, we propose *4D unsupervised object discovery*, jointly discovering objects from 4D data – 3D point clouds and 2D RGB images with temporal information. We present the first practical approach for this task by proposing a *ClusterNet* on 3D point clouds, which is *jointly iteratively optimized* with a 2D localization network. Extensive experiments on the large-scale Waymo Open Dataset suggest that the localization network and ClusterNet achieve competitive performance on both class-agnostic 2D object detection and 3D instance segmentation, bridging the gap between unsupervised methods and full supervised ones. Codes and models will be made available at https://github.com/Robertwyq/LSMOL.

## 1   Introduction

Computer vision researchers have been trying to locate objects in complex scenes without human annotations for a long time. Current supervised methods achieve remarkable performance on 2D detection [31, 15, 30, 38, 6] and 3D detection [49, 27, 34, 33, 47], benefiting from high-capacity models and massive annotated data, but tend to fail for scenarios that lack training data. Therefore, unsupervised object discovery is critical for relieving the demand for training labels in deep networks, where raw data are infinite and cheap, but annotations are limited and expensive.

However, unsupervised object discovery in complex scenes used to believe impractical. Only a few studies pay attention to this field and achieve limited performance in simple scenarios, far inferior to the supervised model. Recent methods [35, 42] discover objects on 2D still-image utilizing the self-supervised learning [7, 43] to distinguish primary objects from the background, then fine-tune a localization network using the pseudo label. Although these methods outperform the previous generation of object proposal methods [39, 2, 50], their detection results are still far behind supervised models. Furthermore, contrastive learning-guided methods have difficulty in distinguishing different instances within the same category. Alternatively, the 3D point cloud can be

36th Conference on Neural Information Processing Systems (NeurIPS 2022).

decomposed into different class-agnostic instances based on proximity cues [5, 4], but due to lack of semantic information, it is difficult to identify the foreground instances. These problems can be mitigated by the complementary characteristics of 2D RGB images and 3D point clouds. The point cloud data provides accurate location information, while the RGB data contains rich texture and color information. Therefore, [37] proposed to aid unsupervised object detection with LiDAR clues, but it still depends on self-supervised models [14] to identify foreground objects. In summary, *all previous methods rely heavily on the self-supervised learning models and overlook the important information from the time dimension*.

To these ends, we propose a new task named *4D unsupervised object discovery*, *discovering objects utilizing 4D data – 3D point clouds and 2D RGB images with temporal information* [25]. The task needs to joint discover objects on RGB images as in *2D Object Detection* and objects on 3D point clouds as in *3D Instance Segmentation*. Thanks to the popularization of LiDAR sensors in autonomous driving and consumer electronics (e.g., iPad Pro), such 4D data has become much more readily available, indicating the great potential of this task for general application.

In this paper, we present the first practical solution for 4D unsupervised object discovery. We proposed a *joint iterative optimization* for *ClusterNet* on 3D point cloud and localization network in RGB images, utilizing the spatio-temporal consistency from multi-modality. Specifically, the ClusterNet was trained with supervision from motion cues initially, which can be obtained from temporally consecutive point clouds. The 3D instance segmentation output by ClusterNet can be further projected to the 2D image as supervision for the localization network. Conversely, 2D detection can also help to refine the 3D instance segmentation by utilizing appearance information. In this way, the 2D localization network and 3D ClusterNet can benefit from each other through joint optimization. Temporal information could serve as a constraint in the optimization.

Our main contributions are as follows: (1) we proposed a new task termed *4D Unsupervised Object Discovery*, aiming at jointly discovering the objects in the 2D image and 3D Point Cloud without manual annotations. (2) we proposed a *ClusterNet* on 3D point clouds for 3D instance segmentation, which is jointly iterative optimized with a 2D localization network. (3) Experiments on the Waymo Open Dataset [36] suggest the feasibility of the task and the effectiveness of our approach. We outperform the state-of-the-art unsupervised object discovery by a significant margin, superior to supervised methods with limited annotations, and even comparable to supervised methods with full annotations.

## 2 Related work

### 2.1 Supervised object detection

**Object detection from Image.** 2D object detection has made great progress in recent years. Two-stage methods represented by the RCNN family [13, 31, 15] extract region proposals first and refine them with deep neural networks. One-stage methods like YOLO [30], SSD [22] and RetinaNet [21] predict the class-wise bounding box in one-shot based on the anchors. FCOS [38] and CenterNet [48] further detect objects without predefined anchors.

**Object detection from Point Cloud.** LiDAR-based 3D object detection develops rapidly along with autonomous driving. Point-based methods [28, 27, 46] directly estimated 3D bounding boxes from point clouds. The computing efficiency is affected by the number of points, so these methods are usually suitable for indoor scenes. Voxel-based methods [49, 45, 33] operate on the 3D voxelized point cloud are capable for large outdoor scenes. However, voxel resolution can greatly affect performance but is limited by computational constraints. Second [45] and PVRCNN [33] further apply sparse 3D convolutions to reduce compute. CenterPoint [47] extends the idea of anchor-free from 2D detection and proposes a center-based representation of bird-eye view (BEV).

### 2.2 Unsupervised object discovery

**Bottom-up clustering**. Clustering methods combine similar elements based on proximity cues, applicable to point clouds and image data. Selective Search [39], MCG [2] and Edge Box [50] can propose a large number of candidate objects with the help of appearance cues, but it is difficult to identify objects from the background. Similarly, point cloud data can decompose into distinct segments according to density-based methods [10, 5, 32] but is unable to determine which is foreground.

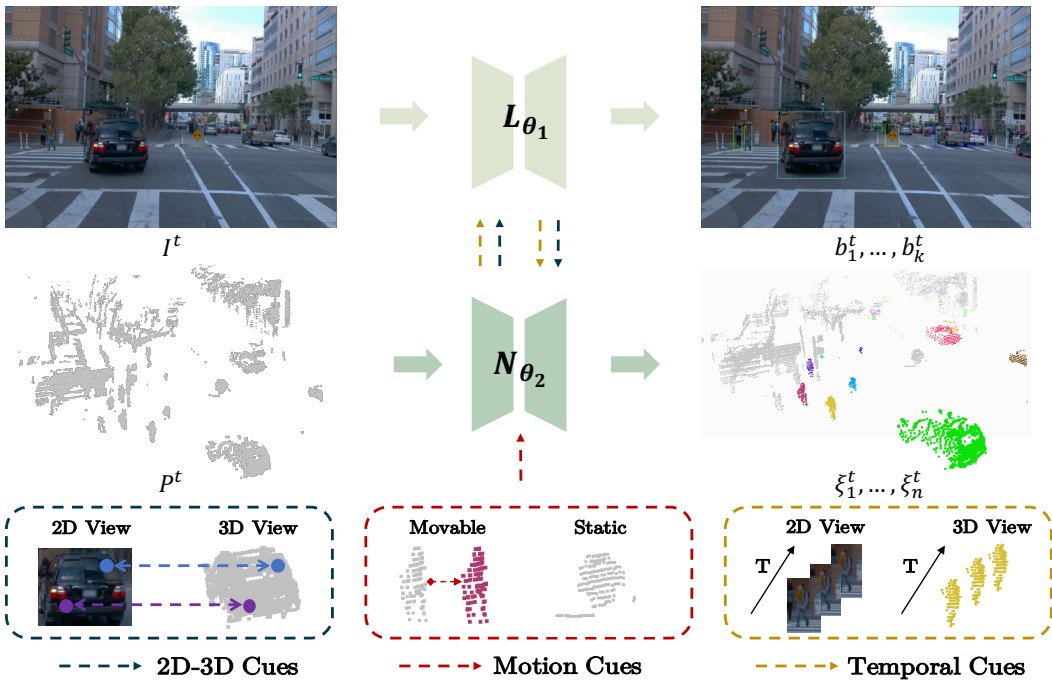

Figure 1: The pipeline of 4D unsupervised object discovery. The input is the corresponding 2D frames and 3D point clouds. The task needs to discover objects on both images and point clouds without manual annotations. The overall process can be divided into two steps: (1) 3D instance initialization and (2) joint iterative optimization. (1) 3D instance initialization: motion cues serve as the initial cues for training the ClusterNet. (2) Joint iterative optimization: the localization network and ClusterNet are optimized jointly by 2D-3D cues and temporal cues.

**Top-down learning**. Recently, self-supervised learning [8, 14, 7, 43] are capable to learn discriminate features without labels. Therefore, many methods attempt to introduce such properties to discriminate foreground objects without manual annotations. LOST [35] utilized a pre-trained DINO [7] to extract primary object attention from the background as the pseudo label and then finetuned an object detector. FreeSOLO [42] further proposed a unified framework for generating pseudo masks and iterative training. However, the performance relies heavily on the pre-trained self-supervised model, which determines the upper limits of such methods. Furthermore, such attention-based methods learned by contrastive learning also have the problem of distinguishing different instances within the same category. Our approach adopts top-down learning as well. Instead of aiding by an external self-supervised model, we look for geometric information to discover objects in the scene naturally.

## 3 Algorithm

### 3.1 Task definition and algorithm overview

The task of *4D Unsupervised Object Discovery* is defined as follows. As shown in Figure 1, the input is a set of video clips recorded in both 2D video frames $I^t$ and 3D point clouds $P^t$ at frame $t$ during training. Since the point cloud and image data provide complementary information about location and appearance, they can serve as the natural cues guiding the training process mutually. During inference, the trained localization network $L_{\theta_1}$ is applied to still-image for 2D object detection, and the trained ClusterNet $N_{\theta_2}$ is applied to the point cloud for 3D instance segmentation.

$$\{b_1^t, ..., b_k^t\} = L_{\theta_1}(I^t), \qquad \{\xi_1^t, ..., \xi_n^t\} = N_{\theta_2}(P^t) \qquad (1)$$

$\{b_1^t, ..., b_k^t\}$ are the 2D bounding box predictions by localization network $L_{\theta_1}$ at frame $t$. $\{\xi_1^t, ..., \xi_n^t\}$ are the 3D instance segments output by ClusterNet $N_{\theta_2}$. $k$ and $n$ denotes the instance index.

$$\theta_1^*, \theta_2^* = \underset{\theta_1, \theta_2}{\arg\min} f(L_{\theta_1}(I^t), N_{\theta_2}(P^t), t) \qquad (2)$$

Our solution exploits the spatio-temporal consistency on 2D video frames and 3D point clouds. The algorithm can be formulated into a joint optimization function $f$ in Eq.2. $\theta_1$ and $\theta_2$ are the parameters of the network need to optimize. Temporal information $t$ serve as the natural constraint in function $f$. The localization network $L_{\theta_1}$ utilized Faster R-CNN as default. We propose a ClusterNet $N_{\theta_2}$ for 3D instance segmentation. Detail implementation will discuss in section 3.2. The major challenge is the optimization for function $f$ without annotations. To overcome the challenge, we seek for *motion cues*, *2D-3D cues* and *temporal cues* to serve as the supervision. All these cues are extracted naturally in the informative 4D data. (1) *motion cues*, represented as 3D scene flow, can distinguish movable segments from the background. It uses to train the ClusterNet $N_{\theta_2}$ initially. (2) *2D-3D cues*, reflecting the mapping between LiDAR points and RGB pixels, can be used as a bridge to optimize the $L_{\theta_1}$ and $N_{\theta_2}$ iteratively. It indicates the output of either network can be further used to optimize another network. (3) *temporal cues*, encouraging the temporal-consistent discovery in 2D and 3D view, can serve as the constraint to optimize the function together. More details will introduce in section 3.3.

## 3.2 ClusterNet

ClusterNet generates 3D instance segmentation from raw point clouds. As shown in Figure 2, given a point cloud $P \in \{(x, y, z)_i, i = 1, ..., N\}$, the network is able to give each point a class type $y_i \in \{1, 0\}$ (indicating foreground or background) and instance ID $d_i \in \{1, ..., n\}$. Thus we can obtain $n$ candidate segments $\xi_i = \{(x, y, z)_j | y_j = 1, d_j = i\}$ on the point cloud. $n$ represents the number of instance segments in one frame of point cloud, and it is different in each frame.

**Network design.** The model first voxelized 3D points $(x, y, z)_i$ and extract voxelized features by a transformer-based feature extractor [11]. We further project these voxelized features back to each point. The feature dimensions of points become $3 + C$ (3 means $XYZ$ and $C$ denotes the embedding dim). Inspired by the VoteNet [27], we leverage a voting module to predict the class type and center offset for each point. Specifically, the voting module is realized with a multi-layer perception (MLP) network. The voting module takes point feature $f_i \in \mathcal{R}^{3+C}$ and outputs the Euclidean space offset $\Delta x_i \in \mathcal{R}^3$ and class type prediction $y_i$. The final loss is the weighted sum of the class prediction and center regression:

$$\mathcal{L} = \mathcal{L}_{center} + \lambda \mathcal{L}_{cls} \tag{3}$$

The class prediction loss $\mathcal{L}_{cls}$ choose the focal loss [21] to balance the points of foreground and background. The predicted 3D offset $\Delta x_i$ is supervised by a regression loss:

$$\mathcal{L}_{center} = \frac{1}{M} \Sigma_i \|\Delta x_i - \Delta x_i^*\|_1 \mathbb{1}[y_i^* = 1] \tag{4}$$

where $\mathbb{1}[y_i^* = 1]$ indicates whether a point belongs to the foreground according to the ground truth $y_i^*$. $M$ is the total number of foreground points. $\Delta x_i^*$ is the ground truth offset from the point position $x_i$ to the instance center it belongs to. According to spatial proximity, we could further group the points into candidate instance segments with the predicted class type and center offset.

**3D instance initialization.** It is more challenging to obtain the supervision signal without manual annotation than the network design. The model was trained initially by motion cues. Specifically, motion provides strong cues for identifying foreground points and grouping parts into objects since moving points generally belong to objects and have the same motion pattern if they belong to the same instance. We could estimate the 3D scene flow $S^t$ from the sequence of point clouds $P^t$ using the unsupervised method [19] at frame $t$. 3D scene flow describes the motion of all the 3D points in the scene, represented as $S^t = \{(v_x, v_y, v_z)_i^t, i = 1, ..., N\}$.

Combining the scene flow $(v_x, v_y, v_z)_i$ and point location in 2D $(u, v)_i$ and 3D $(x, y, z)_i$, we can obtain $(u, v, x, y, z, v_x, v_y, v_z)_i$ for each point $p_i$ in $P^t$. Then, we cluster the points with HDBSCAN [5] to divide the scan into $m$ segments, which will be the instance candidates $\xi_1, ..., \xi_m$. However, these instance candidates contain both foreground and background segments. We further assign each point $p_i$ of segment $\xi_j$ a binary label $y_i^*$ to distinguish foreground points using the motion cues (3D scene flow), as shown in Eq. 5.

$$y_i^* = \mathbb{1}[\{\frac{1}{|\xi_j|} \sum_{p_i \in \xi_j} \mathbb{1}[\|S^t(p_i)\|_2 > \sigma]\} > \eta] \tag{5}$$

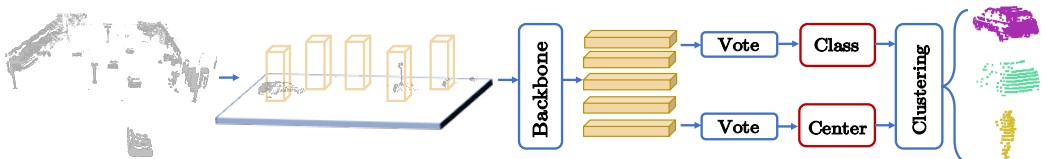

Figure 2: Overview of the ClusterNet architecture. A backbone extracts voxelized features for point clouds, given an input point cloud of $N$ points with $XYZ$ coordinates. Each point predicts a class type and center through a voting module. Then the points are clustered into instance segmentation.

in which $\mathbb{1}[]$ is the indicator function. $|\xi_j|$ represents the total number of points belonging to segment $\xi_j$. $p_i \in \xi_j$ is a point in segment $\xi_j$. $\|S^t(p_i)\|_2$ represents the velocity of the point $p_i$, and $\sigma$ denotes the threshold for velocity. $\eta$ determines the ratio of being a foreground object. $\sigma = 0.05$ and $\eta = 0.8$ by default. $y_i^* = 1$ means the point belongs to foreground segments. When the proportion of moving points in the segment is greater than the threshold $\eta$, we regard it as a foreground object, and all the points it contains are labelled as foreground. These foreground segments selected by motion serve as the pseudo ground truth to train the ClusterNet initially.

### 3.3 Joint iterative optimization

ClusterNet trained by the *motion cues* serves as the initial weights for $\theta_2$, which is the initialization (iter 0) of joint iterative optimization. Although movable objects can separate from the background with the *motion cues*, there are many static objects (e.g., parked cars or pedestrians waiting for traffic lights) in the scenes. Discovering both movable and static objects relies on further joint optimization by *2D-3D cues* and *temporal cues*. In section 3.3.1, we introduce the specific process of joint iterative optimization. Specifically, The 3D segments output by $N_{\theta_2}$ can project to the 2D image to train the $L_{\theta_1}$, and the 2D proposals output by $L_{\theta_1}$ can lift back to 3D view to train the $N_{\theta_2}$. Temporal consistency ensures the objects appear continuously in 2D and 3D views, which is a critical constraint in optimization. The joint optimization can be iterated several times since the 2D localization network and 3D ClusterNet can benefit from each other. In section 3.3.2, we will introduce the technical design for static object discovery.

#### 3.3.1 Model training

In Eq. 6, our goal is to optimize the $\theta_1$ and $\theta_2$ without annotations. $I^t$ and $P^t$ denote the RGB image and point cloud at frame $t$. It is challenging to optimize both parameters simultaneously, so we divide the optimization process into two iterative steps: 2D step and 3D step.

$$\theta_1^*, \theta_2^* = \arg\min_{\theta_1, \theta_2} f(\boldsymbol{L}_{\boldsymbol{\theta_1}}(I^t), \boldsymbol{N}_{\boldsymbol{\theta_2}}(P^t), t)$$

$$\text{2D Step:} \quad \theta_1^* = \arg\min_{\theta_1} f(\boldsymbol{L}_{\boldsymbol{\theta_1}}(I^t), N_{\theta_2}(P^t), t) \tag{6}$$

$$\text{3D Step:} \quad \theta_2^* = \arg\min_{\theta_2} f(L_{\theta_1}(I^t), \boldsymbol{N}_{\boldsymbol{\theta_2}}(P^t), t)$$

**2D step.** In this step, the $\theta_2$ is fixed and optimized $\theta_1$. Since the ClusterNet $N_{\theta_2}$ are able to generate 3D instance segments $\xi_1, ..., \xi_n$ in 3D space, we can further project the 3D instance segments to 2D image plane by the transformation $T_{cl}$ (from the LiDAR sensor to the camera) and projection matrix $P_{pc}$ (from camera to pixels) defined by the camera intrinsic.

$$\begin{pmatrix} \boldsymbol{u} \\ 1 \end{pmatrix} = P_{pc} T_{cl} \begin{pmatrix} \boldsymbol{x} \\ 1 \end{pmatrix} \tag{7}$$

in which $\boldsymbol{u}$ denotes the pixel location in the 2D image plane, and $\boldsymbol{x}$ represents the 3D position of LiDAR points. Hence we can obtain the object point sets $\{\omega_1, .., \omega_n\}$ in the 2D image plane by projecting the LiDAR points of 3D instance segments $\{\xi_1, ..., \xi_n\}$. The 2D bounding boxes $\{b_1^*, ..., b_n^*\}$ derived from projected object point sets $\{\omega_1, .., \omega_n\}$, can use to optimize the weights of localization network $L_{\theta_1}$.

**3D step.** In this step, the $\theta_1$ is fixed and optimized $\theta_2$. The localization network $L_{\theta_1}$ can output 2D bounding box predictions $\{b_1^a, ..., b_k^a\}$ based on the image appearance information. It enables us to discover more objects in the scene (e.g., parked cars regarded as background by motion cues). We can get the updated 2D object set $b^*$ by Eq. 8 (box IoU set to 0.3 in Non-Maximum Suppression).

$$b^* = NMS(\{b_1^*, ..., b_n^*\} \cup \{b_1^a, ..., b_k^a\}) \tag{8}$$

Since many 3D instance segments may have been labelled as background by motion cues before, we later refined the label with the help of the 2D object set $b^*$. Although the projection from LiDAR to the image is non-invertible without dense depth maps, we could still utilize the mapping between the LiDAR point and image pixels. It suggests using the LiDAR points within the 2D bounding box to relabel the 3D instance segments. However, the bounding box may contain many LiDAR points corresponding to different 3D instance segments. Practically, we only consider the primary segment $\xi_j$ (with most points) inside the bounding box and relabel the primary segment as the foreground object. With the refined label, we obtain the updated 3D object set $\xi^*$ of 3D instance segments $\{\xi_1^*, ..., \xi_n^*\}$, which further utilize to optimize the weights for ClusterNet $N_{\theta_2}$.

**Temporal cues.** Temporal information can be integrated into the 2D step and 3D step as extra constraints. As shown in Eq. 9, $b^t$, $\xi^t$ represent the predicted 2D bounding box set and predicted 3D segments set for frame $t$ by $L_{\theta_1}$ and $N_{\theta_2}$. $b_*^t, \xi_*^t$ denote the pseudo annotation for 2D bounding boxes ($\{b_1^*, ..., b_n^*\}$) and 3D instance segments ($\{\xi_1^*, ..., \xi_n^*\}$) from previous 2D step and 3D step. $\mathcal{L}_{2D}$ is the loss for the localization network, and $\mathcal{L}_{3D}$ is the loss for ClusterNet, which is introduced in Eq. 3. $\mathcal{L}_{smooth}$ encourages that the same object has consistent object labels across frames. The constraint can be added to both 2D views and 3D views. Therefore, it can help find new potential objects and filter out wrong annotations across time. More details are illustrated in Appendix C.

$$f(L_{\theta_1}(I^t), N_{\theta_2}(P^t), t) = \underbrace{\mathcal{L}_{2D}(b^t, b_*^t) + \mathcal{L}_{smooth}(b_*^t)}_{\text{2D step}} + \underbrace{\mathcal{L}_{3D}(\xi^t, \xi_*^t) + \mathcal{L}_{smooth}(\xi_*^t)}_{\text{3D step}} \tag{9}$$

### 3.3.2 Static object discovery

Static object discovery is crucial in joint iterative training since the initialization by motion could handle movable objects well. During the joint iterative training, two technical designs are important for static object discovery. One is from the aspect of visual appearance, the other is from the aspect of temporal information.

**Discover static objects by visual appearance.** 2D localization network learns the object representation by visual appearance. It indicates the good generalization ability for static objects since movable objects and static objects usually have similar visual appearances. However, a critical design is the selection of positive and negative samples in model training. Initially, the 2D pseudo annotations generated by motion cues mainly come from moving objects. It is crucial to avoid static objects becoming negative samples so that the model can have better generalization ability to static objects. Table 4 compares different sampling strategies for the training.

**Discover static objects by temporal information.** Temporal information is also beneficial for static object discovery. Due to the occlusion in the 2D view, it is more applicable to discover potential new objects by tracking in the 3D view. Practically, we used Kalman filtering for 3D tracking, and rediscover new objects in the static tracklets (center offset between the start and end frames less than 3 meters). Since we only focus on static objects, the mean center of the tracklet would be a good prediction for lost objects.

## 4 Experiments

### 4.1 Dataset and implementation details

We evaluate our method on the challenging Waymo Open Dataset (WOD) [36], which provides 3D point clouds and 2D RGB image data that is suitable for our task setting. It is of great significance to verify our unsupervised method under such a real and large-scale complex scene.

**Dataset.** Waymo Open Dataset [36] is a recently released large-scale dataset for autonomous driving. We utilize point clouds from the 'top' LiDAR (64 channels, a maximum distance of 75 meters),

and video frames (at a resolution of 1280×1920 pixels) from the 'front' camera. The training and validation sets contain around 158k and 40k frames, respectively. All training frames and validation frames are manually annotated with 2D bounding boxes and 3D bounding boxes, which are capable of evaluating the performance of 2D object detection and 3D instance segmentation. Furthermore, WOD also provides the scene flow annotation in the latest version [17], which can illustrate the upper potential of our method.

**Evaluation protocol.** Evaluation is conducted on the annotated validation set of WOD. We evaluate the performance of 2D object detection and 3D instance segmentation. The dataset contains four annotated object categories ('vehicles', 'pedestrians', 'cyclists', and 'sign'). We test the class-agnostic average precision (AP) score for vehicles, pedestrians, and cyclists. For 2D object detection, the AP score is reported at the box intersection-over-union (IoU) threshold of 0.5. For better analysis, results are also evaluated on small (area $< 32^2$ pixels), medium ($32^2$ pixels $<$ area $< 96^2$ pixels) and large objects (area $> 96^2$ pixels). We also calculated the average recall (AR) to measure the ability of object discovery. For 3D instance segmentation, no previous metrics have been proposed on WOD. Referring to the 2D AP metrics, we propose to compute 3D AP score based on the IoU between predicted instance point sets and the ground truth. The ground truth for the instance segmentation can be obtained by labelling the point within 3D bounding boxes. The 3D AP score is reported at the point sets IoU threshold of 0.7 and 0.9, denoted as $AP^{70}$ and $AP^{90}$, respectively. We also calculated the recall and foreground IoU for better analysis, which can measure the ability of object discovery from more perspectives. Note here the $AP^{2D}$ denotes 2D object detection $AP^{50}$ score. $AP^{3D}$ denotes the 3D instance segmentation $AP^{70}$ score.

**Implementation details.** Our implementation is based on the open-sourced code of mmdetection3d [9] for 3D detection and detectron2 [44] for 2D detection. For 2D localization network, we utilize Faster R-CNN [31] with FPN [20] by default, where ResNet-50 [16] is used as the backbone. The network is trained on 8 GPUs (A100) with 2 images per GPU for 12k iterations. The learning rate is initialized to 0.02 and is divided by 10 at the 6k and the 9k iterations. The weight decay and the momentum parameters are set as $10^{-4}$ and 0.9, respectively. For 3D ClusterNet, the input raw point clouds removed ground points first by [4] and remained the points that can only be seen on the front camera. The cluster range is $[0m, 74.88m]$ for the X-axis, $[-37.44m, 37.44m]$ for the Y-axis and $[-2m, 4m]$ for the Z-axis. The voxel size is $(0.32m, 0.32m, 6m)$. The feature extractor for voxelized points is [11], and the embedding dim for $C$ is 128. In the focal loss for class prediction, we set $\gamma = 2.0, \alpha = 0.8$. The balance weight $\lambda$ for Eq. 3 is set to 5. During inference, we set the minimum number of points to 5 for clustering. The ClusterNet is trained on 8 GPUs (A100) with 2 point clouds per GPU for 12 epochs. The learning rate is initialized to $10^{-5}$ and adopts the cyclic cosine strategy (target learning rate is $10^{-3}$). For hyper-parameters in HDBSCAN [5], we set the min cluster size to 15, and the others follow the default. For more implementation details, please refer to Appendix A.

## 4.2 Main results

**2D object detection.** Table 1 compares the results between annotation from manual and annotaion from our method (termed as ClusterNet) for class-agnostic 2D object detection. All the experiments in Table 1 utilized the same model (Faster R-CNN [31]) for fair evaluation. For the 2D bounding box annotations, the distant boxes are LiDAR invisible. The manual-annotated supervised baseline is trained with LiDAR visible 2D box annotations for a fair comparison. Even though our method still has the gap **43.2 vs. 54.4** to supervised baseline using fully manual annotation (1137k bounding boxes), we can outperform the case when the annotation is limited. Limited annotation is frequent in real-world applications. Compared with 10% manual annotation (127k bounding boxes), our method could achieve 43.2 AP without any manual annotations and beat the 33.8 AP by a large margin. Since our method relies on the motion cues estimated unsupervised, we proved that the performance could increase to an incredible 51.8 AP with ground truth scene flow, which is very *close to the performance of fully manual annotation but without bounding box annotation*. Since the previous unsupervised methods only focused on still 2D images and could not extract the objects from the background accurately, it is no surprise they could only achieve poor results in such challenging scenes. LOST [35] can only extract one primary object from the background, which does not apply to the driving scenes. Freesolo [42] often generates a large mask for a row of cars, which can not distinguish specific instances. Felzenszwalb Segmentation [12] generate potential proposals by graph-based segmentation but lacks the ability to identify the foreground objects. Our method has great performance advantages over these still-image methods.

Table 1: Class-agnostic 2D object detection

| annotation setting | #images | #bboxes | network weights initialized from | $AP^{50}$ | $AP^{50}_S$ | $AP^{50}_M$ | $AP^{50}_L$ | $AR^{50}$ | $AR^{50}_S$ | $AR^{50}_M$ | $AR^{50}_L$ |
|---|---|---|---|---|---|---|---|---|---|---|---|
| *supervised* | | | | | | | | | | | |
| fully manual annotation | 158k | 1137k | ImageNet | 54.4 | 20.5 | 72.4 | 90.9 | 62.8 | 35.5 | 80.8 | 94.0 |
| fully manual annotation | 158k | 1137k | scratch | 52.5 | 23.5 | 67.6 | 86.3 | 62.3 | 34.9 | 80.0 | 93.3 |
| 10% manual annotation | 15k | 127k | ImageNet | 33.8 | 5.5 | 45.3 | 74.9 | 36.1 | 9.7 | 48.6 | 76.7 |
| 10% manual annotation | 15k | 127k | scratch | 31.6 | 5.7 | 42.5 | 72.2 | 35.9 | 8.6 | 47.7 | 75.3 |
| *unsupervised* | | | | | | | | | | | |
| Felzenszwalb [12] | 158k | 0 | ImageNet | 0.4 | 0.0 | 0.5 | 1.1 | 11.1 | 0.6 | 14.5 | 30.7 |
| LOST [35] | 158k | 0 | ImageNet | 1.9 | 0.0 | 1.0 | 7.6 | 5.0 | 0.0 | 0.4 | 27.9 |
| FreeSolo [42] | 158k | 0 | ImageNet | 1.0 | 0.2 | 1.0 | 1.9 | 2.2 | 0.0 | 0.1 | 12.7 |
| **ClusterNet** (w/ gt sceneflow) | 158k | 0 | scratch | 51.8 | 21.3 | 70.2 | 89.5 | 60.8 | 30.2 | 81.2 | 94.8 |
| **ClusterNet** | 158k | 0 | scratch | 43.2 | 18.4 | 56.5 | 81.8 | 55.4 | 26.7 | 71.9 | 93.1 |

**3D instance segmentation.** Table 2 illustrates the effectiveness of our ClusterNet on 3D instance segmentation. Our ClusterNet achieved 26.2 $AP^{70}$ and 19.2 $AP^{90}$ without any annotation, superior to 10% supervised baseline 23.6 $AP^{70}$ and 15.5 $AP^{90}$ with 397k 3D bounding boxes annotation. We proved that our method with accurate motion cues (ground truth scene flow) could achieve 42.0 $AP^{70}$ and 33.2 $AP^{90}$, even comparable to that supervised baseline with fully manual annotation (4268k 3D bounding boxes). No previous method can achieve such high performance under an unsupervised setting. Figure 3 illustrates the object prediction of our approach on the WOD validation set.

Table 2: Class-agnostic 3D instance segmentation

| annotation setting | #point clouds | #3D bboxes | $AP^{70}$ | $AP^{90}$ | $Recall^{70}$ | $Recall^{90}$ | IoU |
|---|---|---|---|---|---|---|---|
| *supervised* | | | | | | | |
| fully manual annotation | 158k | 4268k | 45.7 | 37.3 | 75.1 | 65.1 | 92.2 |
| 10% manual annotation | 15k | 397k | 23.6 | 15.5 | 61.8 | 48.7 | 81.6 |
| *unsupervised* | | | | | | | |
| **ClusterNet** (w/ gt sceneflow) | 158k | 0 | 42.0 | 33.2 | 61.7 | 52.3 | 88.1 |
| **ClusterNet** | 158k | 0 | 26.2 | 19.2 | 40.0 | 32.8 | 64.9 |

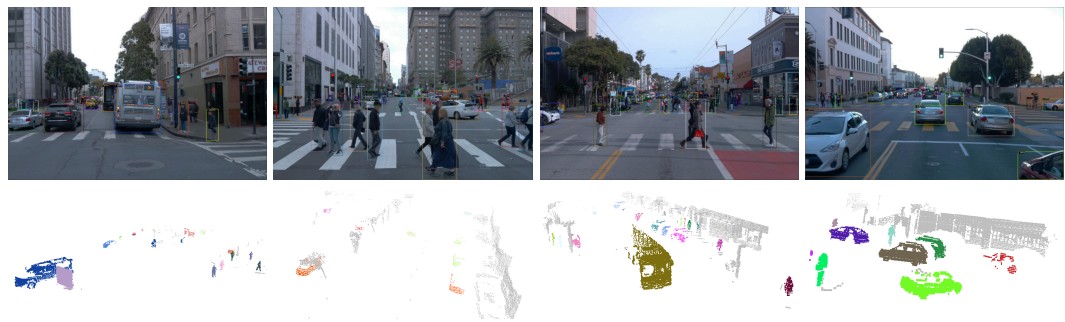

Figure 3: Visualization for 2D object detection and 3D instance segmentation on the WOD validation set. Our approach could achieve such incredible results without any annotations.

**2D instance segmentation.** We can also conduct instance segmentation by projecting the LiDAR points of 3D instance segments to the 2D image plane. The key difference is that instance segmentation masks other than object bounding boxes deriving from 3D instance segments as pseudo annotations. We utilized alpha shape [1] to generate the mask of object points (LiDAR points projected to the 2D image). The localization network can change conveniently to Mask R-CNN [15] for instance segmentation without manual annotations. Some predictions on validation set are illustrated in Figure 4 and Appendix D. Because the Waymo Open Dataset did not provide the annotation for instance mask, the performance of instance segmentation cannot be evaluated quantitatively.

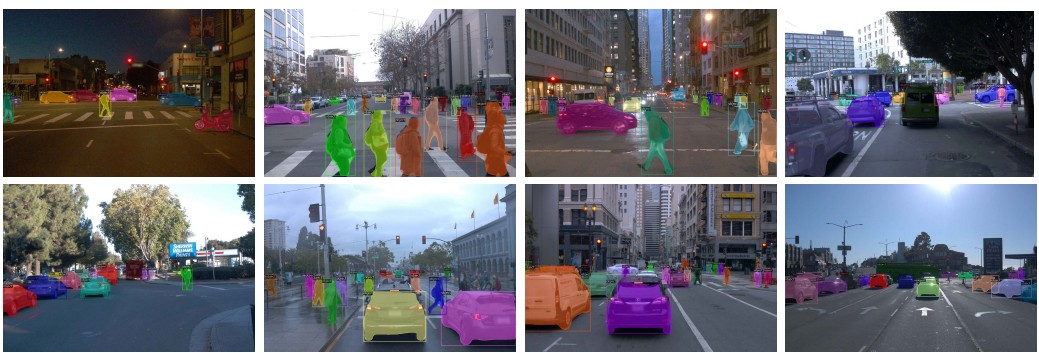

Figure 4: Instance segmentation by our approach when using Mask R-CNN [15] as our localization network, without manual annotations. Our method can generate high-quality instance masks.

## 4.3   Ablation studies

**Analysis of multi cues for training.** Table 3 analyze the contributions of multi cues in our approach to the WOD validation set. The final results are obtained after three iterations of joint optimization. $AP^{2D}$ denotes the $AP^{50}$ score for 2D object detection, and $AP^{3D}$ denotes the $AP^{70}$ score for 3D instance segmentation. The ClusterNet was trained with the pseudo-annotations obtained by HDBSCAN Clustering [5] for the first time. So a simple baseline is directly using the HDBSCAN for 3D instance segmentation and project on 2D for localization network training. In comparison, we demonstrate the effectiveness of using ClusterNet; the performance increased by 4.9 $AP^{2D}$ (from 25.1 to 30.0) and 2.2 $AP^{3D}$ (from 4.6 to 6.8). Furthermore, the performance improved significantly by joint optimizing for the ClusterNet and localization network, with 2D-3D cues and temporal cues.

Table 3: Analysis of multi cues for training.

| Method | point cloud | motion cues | 2D-3D cues | temporal cues | $AP^{2D}$ ↑ | $AP^{3D}$ ↑ |
|---|---|---|---|---|---|---|
| HDBSCAN [5] | ✓ | | | | 14.9 | 2.1 |
| | ✓ | ✓ | | | 25.1 | 4.6 |
| ClusterNet | ✓ | ✓ | | | 30.0 | 6.8 |
| | ✓ | ✓ | ✓ | | 40.4 | 25.7 |
| | ✓ | ✓ | ✓ | ✓ | **43.2** | **26.2** |

Table 4: Ablation on sampling strategy.

| sampling strategy | $AP^{50}$ | $AP^{50}_S$ | $AP^{50}_M$ | $AP^{50}_L$ |
|---|---|---|---|---|
| (a) IoU$_+$>0.7, IoU$_-$<0.3 | 27.8 | 3.2 | 37.2 | 70.0 |
| (b) IoU$_+$>0.6, IoU$_-$<0.4 | 28.2 | 3.3 | 36.9 | 71.7 |
| (c) **IoU$_+$>0.6, 0.1<IoU$_-$<0.4** | **30.0** | **4.3** | **39.4** | **73.2** |

Table 5: Ablation on early stopping.

| iterations | $AP^{50}$ | $AR^{50}$ |
|---|---|---|
| **3000** | **30.0** | **43.3** |
| 6000 | 27.9 | 40.4 |
| 12000 | 25.3 | 35.0 |

**Training strategy for localization network.** Training the localization network with the pseudo annotations generated by ClusterNet is quite different from training with manual annotations. Specifically, the pseudo annotations will be noisy and incomplete before joint iterative training. Therefore, two points we found crucial for the initial training: (1) sampling strategy in RPN (Region Proposal Network) and (2) early stopping for training. The following ablation experiments are conducted for the first-time training of the localization network. Table 4 compares three different strategies: (a) sample anchors with box IoU > 0.7 as positive example and box IoU < 0.3 as negative example, as in the standard Faster R-CNN; (b) sample anchors with box IoU > 0.6 as positive example and box IoU < 0.4 as negative example; (c) sample anchors with box IoU > 0.6 as positive example, and box 0.1 < IoU < 0.4 as negative example. In this way, the strategy considerably reduces the chance

of sampling static objects as negative examples. Table 5 compares different training iterations and shows that early stopping improves the generalization performance. Since the pseudo annotations have noise, training for a long time may overfit the noise in the training set, leading to the degradation of generalization performance.

**Joint iterative optimization.** Table 6 presents the effectiveness of our joint iterative optimization for the 2D localization network and 3D ClusterNet. Iteration 0 represents the initial performance of ClusterNet trained by motion cues (estimated scene flow). Next, each iteration means a 2D step and a 3D step. Even though the model did not perform well at the beginning, with joint iterative optimization, both $AP^{2D}$ and $AP^{3D}$ improved rapidly. Applying more than one iteration improves the results, indicating that the 2D localization network and 3D ClusterNet can benefit from each other. We set the iteration number as 3 by default.

**Minimum points for ClusterNet.** Minimum points determine the minimum number of LiDAR points for 3D instance segments. Table 7 analyze the model performance under different parameters during the inference. We set the min points to 5 by default.

Table 6: Joint iterative optimization.

| iterations | $AP^{2D}$ | $AP^{3D}$ |
|:---:|:---:|:---:|
| 0 | / | 6.8 |
| 1 | 30.0 | 20.2 |
| 2 | 37.4 | 25.4 |
| **3** | **43.2** | **26.2** |
| 4 | 42.8 | 25.9 |

Table 7: Minimum points for ClusterNet.

| min points | $AP^{3D}$ |
|:---:|:---:|
| 2 | 25.3 |
| **5** | **26.2** |
| 10 | 26.0 |
| 20 | 25.5 |

## 5 Discussion and conclusions

**Discussion.** Unsupervised object discovery used to believe infeasible due to the ambiguity of objects and the complexity of scenarios. However, 4D data with the sequence of image frames and point clouds provide enough cues to discover the movable objects, even without manual annotation. The complementary information behind the 3D LiDAR points and 2D image and constraints from temporal are the critical factors for the success of unsupervised object discovery. With 4D sensor data readily available onboard, our approach shows extraordinary potential for scenarios with limited or no annotation. The only limitation is that our method is suitable for movable objects (vehicles, pedestrians); static things (never move) like beds or chairs can not be discovered.

**Conclusions.** In this work, we propose a new task named *4D Unsupervised Object Discovery*. The task needs to discover objects both on the image and point clouds without manual annotations. We present the first practical approach for this task by proposing a *ClusterNet* for 3D instance segmentation and joint iterative optimization. Extensive experiments on the large-scale Waymo Open Dataset demonstrate the effectiveness of our approach. So far as we know, we are the first work to achieve such high performance for unsupervised 2D object detection and 3D instance segmentation, bridging the gap between unsupervised methods and supervised methods. Our work sheds light on a new perspective on the future study of unsupervised object discovery.

**Societal Impacts.** The development of unsupervised object discovery requires large datasets, introducing privacy issues. The technology of unsupervised detection dramatically reduces the labelling cost; it may affect the people engaged in the labelling industry in the future. The elimination of human intervention may also cause some data annotators to lose their current jobs. Our approach only tests in driving scenes for effectiveness, which may lead to some wrong detection in other scenes.

## Acknowledgments and Disclosure of Funding

The authors thank the anonymous reviewers for their constructive comments. This work was supported in part by the Major Project for New Generation of AI (No.2018AAA0100400), the National Natural Science Foundation of China (No. 61836014, No. U21B2042, No. 62072457, No. 62006231), and the InnoHK program. The authors would like to thank Xizhou Zhu and Jifeng Dai for conceiving an early idea of this work. Also, our sincere and hearty appreciations go to Jiawei He, Lue Fan and Yuxi Wang, who polishes our paper and offers many valuable suggestions.

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
