# APPENDIX

## A    Implementation details

**Scene flow estimation.** Scene flow records the motion of all the 3D points in the scene. In our task, we estimate the 3D scene flow of LiDAR points as *motion cues*. We need to pre-process the point cloud data first. Specifically, we remove the ground points of raw point clouds by [4] and transform the points into global coordinates. Taking two consecutive point clouds ($P^t$, $P^{t+1}$) as the input, we use the method [19] to estimate the 3D scene flow $S^t$ of the current frame $P^t$. The hyper-parameters are briefly presented here. We use all the points after pre-processing. The batch size is set to 1, and the learning rate is 0.001. We use Adam [18] for the optimization, and the max iterations set to 5000. The hidden units are 128, and the early patience we adjust to 300. More details can be found in [19].

**Training for localization network.** We utilize Faster R-CNN [31] as the localization network. For all experiments, the choice of hyper-parameters for Faster R-CNN follows the latest Detectron2 [44] code base. Models are trained on images of shorter side {480, 512, 544, 576, 608, 640, 672, 704, 736, 768, 800} pixels. Random cropping is used for data augmentation during training, and the relative range is (0.9,0.9). In inference, the shorter side is 1280 pixels. Anchors are of 5 scales and 3 aspect ratios. 1k region proposals are generated at an NMS threshold of 0.7. During inference, detection results are derived at an NMS threshold of 0.5, score threshold of 0.01. 256 anchor boxes (at a positive-negative ratio of 1:1) and 256 region proposals (at a positive-negative ratio of 1:3) are sampled for RPN and Faster R-CNN training, respectively. The networks are trained on 8 GPUs (NVIDIA A100) with 2 images per GPU. As mentioned in the ablation study, we only train for 3k iterations for the initial training. The learning rate is initialized to 0.02 and is divided by 10 at the 1.5k and 2.5k iterations. For other training processes, we train for 12k iterations. The learning rate is initialized to 0.02 and is divided by 10 at the 6k and the 9k iterations. We iterated the joint training three times, as mentioned in the ablation study. The training process only takes two hours for one iteration.

**Training for ClusterNet.** We use 4 sparse regional attention blocks of SST [11] as the voxel feature extractor and keep voxelization parameters the same as [11]. The input raw point clouds removed ground points first by [4] and remained the points that can only be seen on the front camera. The cluster range is $[0m, 74.88m]$ for the X-axis, $[-37.44m, 37.44m]$ for the Y-axis and $[-2m, 4m]$ for the Z-axis. The voxel size is $(0.32m, 0.32m, 6m)$. Except for the frame having nearly no scene flow estimation, we use the remained frames of point clouds in the Waymo Open Dataset training set to train the model. It contains 125415 frames for training. The ClusterNet is trained on 8 GPUs (NVIDIA A100) with 2 point clouds per GPU for 12 epochs. The learning rate is initialized to $10^{-5}$ and adopts the cyclic cosine strategy (target learning rate is $10^{-3}$). The training process takes 8 hours. During inference, we use the *Connected Component* (CC) algorithm for instance grouping. All the points voted as the foreground are used as vertices in the graph. Two vertices belong to the same instance if their distance is smaller than a certain threshold. The threshold is set to 0.6 in our experiment setting.

## B    Other methods on Waymo Open Dataset

**LOST [35].** We follow the code base released by [35] and directly apply the model to Waymo Open Dataset (WOD). We use the ViT-S model introduced in [7], which is trained using DINO [7]. With a set of unlabeled images, LOST only can extract one bounding box for a prominent object per image. Then we train our localization network with the pseudo labels generated by LOST. The training strategy is the same as the initial training in A.

**Felzenszwalb Segmentation [12].** The algorithm can segment an image into regions by graph-based representation. The hyper-parameters we used on Waymo Open Dataset (WOD) are listed below. The scale is 100, the sigma is 10, and min size is 500. It is challenging to find a good pair of parameters for diverse scenes in WOD. Besides, the method is unable to distinguish foreground objects from the background, which leads to poor accuracy in the results.

**HDBSCAN [5].** In the ablation study, we compare the results of 3D instance segmentation directly using the HDBSCAN algorithm [5]. The min cluster size is set to 15 and the others follow the default. We estimated the 3D scene flow of each frame $P^t$ on the validation set. Combining the scene flow

$(v_x, v_y, v_z)_i$ and point location in 2D $(u, v)_i$ and 3D $(x, y, z)_i$, we can obtain $(u, v, x, y, z, v_x, v_y, v_z)_i$ for each point $p_i$ in the point cloud $P^t$. Then, we cluster the points with HDBSCAN [5] to divide the scan into several segments as the result of 3D instance segmentation. The min cluster size is set to 5 (less than 5 points will be filtered out), the same as the parameter in ClusterNet. Then we calculate $AP^{3D}$ with the ground truth (points within 3D annotated boxes). Without motion cues, the algorithm can also output the 3D segments, but mixed with the background.

**Comparison to State-of-the-art Methods.** Table 8 compares the results of our *ClusterNet* and other state-of-the-art methods for 2D Class-agnostic Object Detection. LOST [35], Felzenszwalb Segmentation [12] and MCG [2] are the methods applied to still-image. The tasks of zero-shot video object segmentation and motion segmentation are also relevant. We tried to generate annotations by state-of-the-art approaches for video object segmentation (COSNet [23] following the DAVIS-2016 [24], and RVOS [40] following the DAVIS-2017 [26]), and motion segmentation (FuseMODNet [29], using LiDAR). These approaches *are pre-trained with manual annotations* following their original papers. They produce object labels on the Waymo training set. Because COSNet and FuseMODNet generate binary masks without differentiating individual instances, we regard each connected component of the binary masks as an object instance. These approaches show inferior results to our method, even though they are pre-trained with manual annotations using other datasets. Our method outperforms the state-of-the-art unsupervised object discovery by a significant margin, achieving **43.2 AP** score.

Table 8: **Comparison to State-of-the-art Methods for 2D Class-agnostic Object Detection**. † [23], [40] and [29] are pre-trained with manual annotations following their original papers.

| annotation setting | #images | #bboxes | $AP^{50}$ | $AP_S^{50}$ | $AP_M^{50}$ | $AP_L^{50}$ |
|---|---|---|---|---|---|---|
| *still-image* | | | | | | |
| Felzenszwalb Segmentation [12] | 158k | 0 | 0.4 | 0.0 | 0.5 | 1.1 |
| LOST [35] | 158k | 0 | 1.9 | 0.0 | 1.0 | 7.6 |
| MCG [2] | 158k | 0 | 6.9 | 0.0 | 3.0 | 37.5 |
| *zero-shot video* | | | | | | |
| †video object segmentation [23] | 158k | 0 | 10.3 | 0.0 | 2.5 | 45.4 |
| †video object segmentation [40] | 158k | 0 | 14.3 | 0.0 | 6.6 | 54.1 |
| †motion segmentation [29] | 158k | 0 | 24.9 | 3.7 | 29.6 | 58.1 |
| *4D data* | | | | | | |
| **ClusterNet** | 158k | 0 | 43.2 | 18.4 | 56.5 | 81.8 |

# C    Loss for temporal cues

Temporal consistency encourages the objects to be discovered continuously in 2D view and 3D view. In 2D view, objects are represented as a set of bounding boxes $\{b_i^t\}_{i=1}^n$ at frame $t$. In 3D view, objects are represented as a set of segments $\{\xi_j^t\}_{j=1}^m$ at frame $t$. $n$ and $m$ are the max numbers of the objects in the 2D view and 3D view, respectively. Since the definitions of $\mathcal{L}_{smooth}$ in 2D and 3D are similar here, we only explain the case of 2D in the following. Each object has its corresponding label $y_i^t$, whether in 2D view or 3D view. $y_i^t = 1$ denotes the foreground object while $y_i^t = 0$ means background. $t$ is the index of the frame.

The core idea of temporal cues is to constrain the consistency of labels $\{y_i^t\}_{i=1}^n$, because an object cannot have different labels across frames. Therefore, under the constraint of time consistency, we can *filter out the wrong foreground estimation* (improve precision) and *rediscover new foreground objects from the background* (improve recall). The definition of $\mathcal{L}_{smooth}$ is shown in Eq. 10.

$$L_{\text{smooth}}(y_i^t) = \sum_{j \in \Omega_i^{t \to t+1}} \mathbf{1}\left[y_i^t \neq y_j^{t+1}\right] + \sum_{j \in \Omega_i^{t \to t-1}} \mathbf{1}\left[y_i^t \neq y_j^{t-1}\right]$$
$$\Omega_i^{t \to t+1} = \{j | \text{IoU}_{\text{box}}(b_i^{t \to t+1}, b_j^{t+1}) > \eta_{\text{propagate}}\}$$

(10)

$\Omega_i^{t \to t+1}$ is the set of object indexes at the $t + 1$-th frame, which highly overlap with the object set $\{b_i^t\}_{i=1}^n$ (with the cross-frame motion taking into account). Practically, Kalman filtering is used for

motion estimation on 2D by [3] and on 3D by [41]. $b_i^{t \to t+1}$ is derived by propagating object $b_i^t$ at time $t$ to time $t+1$ according to the motion estimation by Kalman filtering. If the bounding boxes of $b_i^{t \to t+1}$ and $b_j^{t+1}$ have an overlap large than $\eta_{propagate}$, they are much likely to be of the same object. The loss term will penalize if they have different labels. This term is also applied along the inverse time axis. $\Omega_i^{t \to t-1}$ takes the same formulation as $\Omega_i^{t \to t+1}$ at inverted axis. In 2D view, $\eta_{propagate} = 0.3$ due to the scale projection of image plane. However in 3D view, one difference is $\eta_{propagate} = 0.9$. Another is 2D bounding box replaced by 3D bounding box derived from the segments $\{\xi_j^t\}_{j=1}^m$ to compute IoU (Intersection over Union). We filter out tracklets with a length of less than 5. These objects are considered to be incorrectly estimated foreground. For the other tracklets, we can rediscover new foreground objects from the background with the help of $\mathcal{L}_{smooth}$.

## D   2D instance segmentation visualization

Figure 5 shows the generated instance segmentation masks in diverse scenes on Waymo Open Dataset.

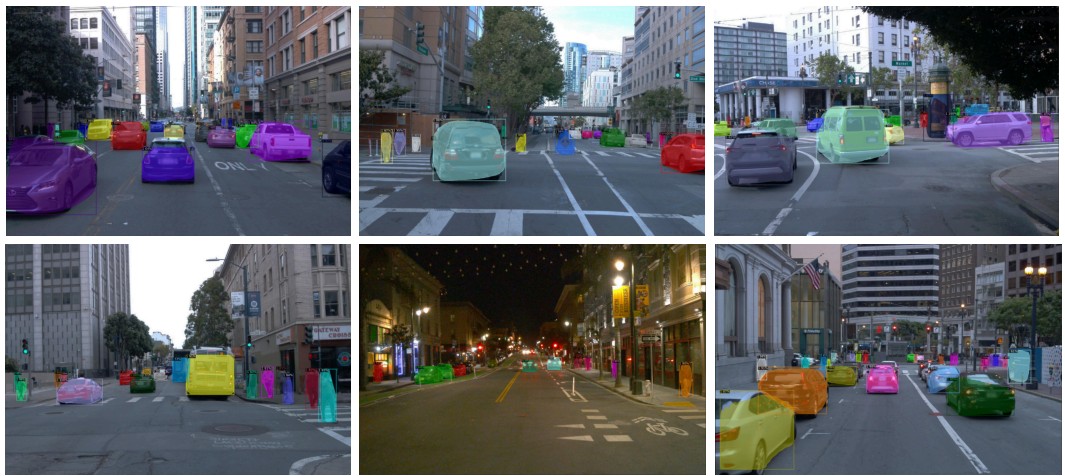

Figure 5: Instance segmentation on Waymo Open Dataset.

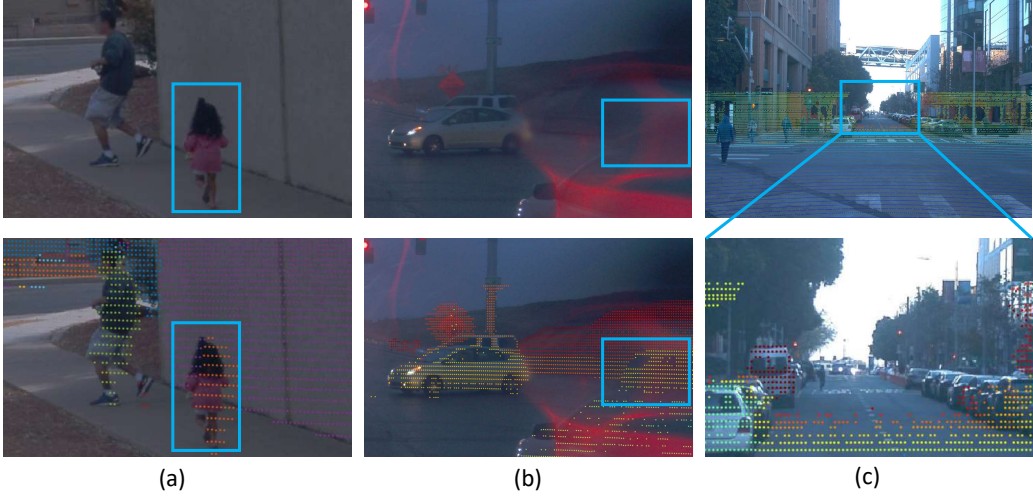

Figure 6: Illustration of known issues: (a) misalignment between image and projected 3D points; (b) invisible car in image because of blur and reflection; (c) distant objects not detectable by LiDAR.

# E  Known issues

A known issue when developing this approach is the inconsistency of the image content and the point clouds. In our experiment, the point clouds are collected by LiDAR while the camera records the images. Figure 6 shows some typical examples of inconsistency. In Figure 6(a), due to the rolling shutter effect of the LiDAR, the point clouds cannot be exactly aligned with the image content in time-space. In Figure 6(b), when it rains, image quality suffers from blur and reflection. In Figure 6(c), the distant small objects are not reachable by LiDAR. These inconsistencies would incur errors in our algorithm and limit the effective range of object discovery.

We hope these issues can be mitigated with the development of visual sensors. In the future, the developing ToF sensors and solid-state LiDAR devices may well produce point clouds more consistent with the 2D images. And we expect that our algorithm can further benefit from the progress.