# OpenReview forum: "4D Unsupervised Object Discovery"
_NeurIPS.cc/2022/Conference — NeurIPS 2022 Accept_

### Official Review · Reviewer_Nejz · 2022-07-02

**Rating:** 6
**Confidence:** 4
**Soundness:** 3 good
**Presentation:** 2 fair
**Contribution:** 3 good

**Summary:**

In this paper, the authors have proposed a novel task named *4D Unsupervised Object Discovery*, aiming at discovery the objects from 4D sequential data (3D point clouds + 2D RGB images), without any human annotation.

For this purpose, they proposed a method called *ClusterNet* to automatically discovery the objects in 2D images (bounding boxes) and the objects in 3D point clouds (point-wise segments), which can be jointly optimized on the 4D sequential data by taking the temporal information into account.

The generated labels are evaluated in 2D object detection and 3D instance segmentation, showing on par results with supervised methods and superior performances compared to other object discovery methods. Furthermore, the 2D object boxes can be converted to 2D object segments by projecting the 3D instance segments to 2D images, which allows for joint segmentation on 4D data.

**Questions:**

1. Since one important objective of object discovery is to reduce human effort by automatically generating object labels, retraining existing models with the generated object labels should be added to show the true efficiency of the proposed object discovery method. The performance gap between models trained with generated labels and human labels can serve as a key indicator for the final application in real world.

2. As this method works on 4D data, the authors propose to initial object labels on 3D, then project to 2D and iteratively optimize them, I am wondering is it possible to initial object labels on both 2D and 3D by clustering moving objects with optical flows and scene flows. This is not necessary, but we might get better initial labels by looking at both modes.

3. The current version of Sec 3.3 is a bit too abstractive, with many details hiding in several parts of the paper.
For example, it is not trivial to train the 2D localization network to discover static objects with initial proposals projected from 3D point clouds, a summarization of the training strategy (L276-290) could be added in Sec 3.3 to help the readers understand quickly how this was done.

**Limitations:**

Yes, the limitations are discussed in the last section of the paper.

**Strengths And Weaknesses:**

Strengths:

* This paper proposes a novel task and a novel method with clear presentations, and thorough ablation studies supporting the validation of each component of the proposed method.

* While existing object discovery works mainly focus at extracting discriminative information from pure 2D or 3D data, this work chooses to combine the best of both modes by leveraging the 4D sequential data, and to consider the temporal information as well as the multi-modality constraints. This removes the dependence of a strong self-supervised learning models, and compensate the weaknesses of each side.

* The superior experimental results shows that the advantages of object discovery on 4D data compared to pure 2D or 3D.

Weaknesses:

* No experimental result was given to demonstrate the quality of discovered objects by training existing models with the generated labels instead of the human annotations.

---

> ### Author Response · Authors · 2022-08-02
> **Response to the comments from Reviewer Nejz**
>
> Thanks for your valuable comments. Appreciation for the approval and constructive suggestions.
>
> Q1: Thanks for this. We agree that it is important for object discovery to reduce human effort by automatically generating object labels. We attempt to evaluate the quality of discovered objects in Table 1. Both the supervised baseline and our localization network utilized the architecture of Faster R-CNN. “ClusterNet” here represents our method to generate object labels. Using our generated object labels as supervision, the Faster R-CNN can achieve 43.2 AP (the model is trained from scratch). Using 1137k human box annotations, the Faster R-CNN can achieve 54.4 AP. Especially when the prediction of motion information is more accurate, we will get better results (51.8 AP). We believe our method is very practical in some scenes with limited annotation. 43.2 AP vs. 33.8 AP when using 127k annotations. Also, in Table 2, the model architecture is the same, both utilized ClusterNet. The difference is the annotation setting (our generated object labels vs. human manual annotations).
>
> Q2: Your suggestion is inspiring to us. To be honest, we have tried to obtain 2D object labels from optical flow before; it requires motion decouple. Since optical flow represents the 2D projection of 3D motion (Ego motion and Object motion), the object motion obtained in this way (optical flow minus ego-motion projection) would be noisy. It could achieve 32.6 AP (initialized from 2D optical flow) compared with 43.2 AP (Now initialized from 3D scene flow). Another reason we use 3D scene flow now is that 2D optical flow usually needs a pre-trained model, but we can get 3D scene flow using non-learning methods. But your suggestion is enlightening; we haven’t tried to combine the two together. We believe a better combination would help initialize robust object labels.
>
> Q3: Thanks for this. We agree that static object discovery is crucial in joint iterative training. We appreciate your suggestions and will add another section (Static Object Discovery) in Sec 3.3 to introduce the 2D training strategy as well as our design for static objects. The outline is as follows:
> 1. Discover static objects by appearance; this part will introduce the training strategy of the 2D localization network. It leveraged the generalization ability of the network to find more potential static objects.
> 2. Discover static objects by temporal cues; this part will introduce how to use tracking to find static objects. The objects that have been found in some frames will be expanded to other frames.

---

### Official Review · Reviewer_WTqt · 2022-07-10

**Rating:** 6
**Confidence:** 4
**Soundness:** 3 good
**Presentation:** 3 good
**Contribution:** 3 good

**Summary:**

The paper solves the problem of unsupervised object discovery by utilizing the 4D information (cloud point, and time), they designed an iterative refinement frameworks which do the 2D information refinement and 4D information refinement one-by-one (usually 4~5 iters).

The results achieves the best result under the new setting. A few ablations about the importance of cues, training strategies, etc are provided to help us to better understand the task and the methods.

**Questions:**

Listed as above.

**Limitations:**

Yes.

**Strengths And Weaknesses:**

Pro:

- Interesting and sound setting to explore the 4d unsupervised object discovery.
- The writing is easy to follow from intuition, to method, and finally experimental parts.


Con:
- Figure 1 is not clear about the dataflow. I am totally missed.
- One detail I am curious is that whether your network is initialized from scratch, or from another pretrained network. That counts about whether you can name it as 'unsupervised'.
- The contribution claim of clusternet, can the author highlight the difference of it with VoteNet?

---

> ### Author Response · Authors · 2022-08-02
> **Response to the comments from Reviewer WTqt**
>
> Thanks for your valuable comments. Appreciation for the approval and constructive suggestions.
>
> Q1: Fig. 1 only shows the high-level data flow. It will be better understood to watch with Table 6. The overall process can be divided into two steps: (1) 3D instance initialization and (2) joint iterative optimization.
> (1) 3D instance Initialization: This step represents iteration 0 in Table 6. It trained ClusterNet with the initial 3D pseudo labels, which utilize motion cues (3D scene flow learned from unsupervised method) to select moving HDBSCAN’s 3D proposals. Details can refer to L133-L150.
> (2) Joint iterative optimization: This step represents iterations 1-4 in Table 6. It iteratively trained our 2D localization network and our 3D ClusterNet by utilizing complementary informations of the two modalities. Take iteration 1 for example; the 3D proposals output by ClusterNet can be projected to 2D pseudo labels, we then trained the 2D localization network with these 2D pseudo labels. This is the 2D step in Sec 3.3. Next, the 2D proposals output by the 2D localization network can be used to refine 3D pseudo labels (using the 2D-3D cues and temporal cues), we subsequently train our ClusterNet with the refined 3D pseudo labels. This is the 3aD step in Sec 3.3. An iteration ends after a 2D step and a 3D step. The joint iterative optimization is consisted of several iterations. During inference, the localization network output 2D object bounding boxes on still-image and the ClusterNet output 3D instance segmentation on point clouds.
>
> Q2: Our network is initialized from scratch without any pre-trained weights. We add a column in Table 1 to specify how the network weights are initialized.
> |  annotation setting   | #images  | #bboxes | network weights initialized from | AP$^{50}$ | AP$^{50}_{S}$ | AP$^{50}_{M}$ | AP$^{50}_{L}$ | AR$^{50}$ | AR$^{50}_{S}$ | AR$^{50}_{M}$ | AR$^{50}_{L}$ |
> |  ----  | ----  | ----  | ----  |----  | ----  |----  | ----  |----  | ----  |----  | ----  |
> |  fully manual annotation  | 158k  |  1137k  |ImageNet |54.4  |20.5 |72.4  |90.9  |62.8  |35.5  |80.8 |94.0  |
> |  fully manual annotation  | 158k  |  1137k  |scratch |52.5  |23.5 |67.6 |86.3  |62.3  |34.9  |80.0 |93.3  |
> | 10% manual annotation  | 15k  |  127k  |ImageNet |33.8  |5.5 |45.3  |74.9  |36.1  |9.7  |48.6 |76.7  |
> | 10% manual annotation  | 15k  |  127k  |scratch |31.6 |5.7 |42.5 |72.2  |35.9  |8.6  |47.7 |75.3  |
> | ClusterNet (w/ gt sceneflow)  | 158k  |  0  |scratch |51.8 |21.3 |70.2 |89.5  |60.8  |30.2  |81.2 |94.8  |
> | ClusterNet  | 158k  |  0  |scratch |43.2 |18.4 |56.5|81.8 |55.4  |26.7  |71.9 |93.1  |
>
> Q3: We borrow the idea of center voting from VoteNet. The main differences are as follows:
> (1) Our network architecture is different. VoteNet uses a point-based backbone(PointNet) for feature extraction while our ClusterNet utilize a voxel-based backbone(Single-stride Transformer). Details can refer to L26-L38 in Appendix A. While VoteNet needs to downsample the whole scene into a fixed number of points, our method can handle more general scenes(no limit on the number of points). This advantage is suitable for our unsupervised object label generation since we need to assign each point an instance id.
> (2) The purposes of voting are different. VoteNet uses the voting module to aggregate features for later 3D object detection while we use voting for the direct 3D instance segmentation.

---

### Official Review · Reviewer_nTyt · 2022-07-11

**Rating:** 5
**Confidence:** 3
**Soundness:** 3 good
**Presentation:** 3 good
**Contribution:** 3 good

**Summary:**

The paper tackles the problem of object discovery. They propose the 4d unsupervised object discovery from 4D data - 3D point clouds and 2D RGB images with temporal information together with a new model, ClusterNet that is jointly iterative and optimized with a 2D localization network. Finally, results are reported on the Waymo Open Dataset.

**Questions:**

1. How is "n" chosen for the number of instances?

2. How hard is to collect data with cloud points?

3. Can you give more details about the fully supervised baseline? Does it use a previously published method?

4. In the implementation details (lines 218-232) it is stated that for example for 2D localization network, Faster R-CNN is utilized. Does this mean that you also initialize some weights with a pre-trained Faster R-CNN model or that you rely on the architecture? This aspect needs to be clearly stated.

**Limitations:**

The authors discuss some of the limitations and some potential societal impact problems.


**Strengths And Weaknesses:**

The paper tackles an important problem. It proposes the task of object discovery from 3D point clouds together with 2D images along with the temporal information which can be of interest for the community. I find the joint optimization to be interesting.

My main concern related to this paper is the comparison with state of the art which I find to be quite limited. I would have wished to see more comparisons with other methods for 2D unsupervised object detection. Since the comparison is quite limited, it is hard to estimate the gain brought by the proposed approach. So, the only estimate that remains is the comparison with the supervised case. However, for this setup I feel like there are little details about the used architecture.

---

> ### Author Response · Authors · 2022-08-02
> **Response to the comments from Reviewer nTyt**
>
> Thanks for your valuable comments. Appreciation for the approval and constructive suggestions.
>
> Q1: We want to add some details to make the description more clear. Here “n” represents the number of instance segments in one frame of point cloud. 1) “n” can be different in each frame; it is not a fixed number. 2) the trained ClusterNet could output arbitrary number of clusters based on the predicted center offset of each point during inference. We use the Connected Component (CC) algorithm for instance grouping. Specifically, we move each point to its corresponding center according to the predicted offset and then group different centers within a distance threshold. Details can be found in Appendix A L35-L38. 3) for initial pseudo label generation, “n” is determined by motion cues. Referring to L139-L150, we select the movable (By Eq.5) segments from HDBSCAN’s cluster results as the initial instance segments.
>
> Q2: Thanks to the rapid development of autonomous driving technology and consumer electronics hardwares, it is more and more convenient to collect synchronized images and point cloud sequences nowadays. Well-synchronized camera-LiDAR systems are widespread in autonomous driving (L4), ADAS applications (L2), and consumer electronics (iPhone and iPad Pro). In the meantime, more and more datasets with well-synchronized camera-LiDAR sequences are generously provided by autonomous driving companies (e.g., Waymo, nuScenes, Argoverse), which make our research topic possible.
>
> Q3: We want to provide some details to make the description more clear. For a fair comparison between our generated labels and the fully supervised baseline, we use the same object detector(Faster R-CNN) and training techniques. We used the code implementation and training recipes provided by Detectron2(https://github.com/facebookresearch/detectron2).
>
> Q4: We would like to thank the reviewer for their kind suggestion. In order to be fully unsupervised, our model did not use any pre-training weights. In Table1, our result 43.2 AP is a Faster R-CNN trained from scratch with pseudo labels generated by our method. For the supervised baseline 54.4 AP, it is trained from ImageNet pre-training. We would make this more clear in Table 1 in the future revision by specifying the pre-training weights in use. We also added two experiments for the fully supervised supervised baseline trained from scratch.
> |  annotation setting   | #images  | #bboxes | network weights initialized from | AP$^{50}$ | AP$^{50}_{S}$ | AP$^{50}_{M}$ | AP$^{50}_{L}$ | AR$^{50}$ | AR$^{50}_{S}$ | AR$^{50}_{M}$ | AR$^{50}_{L}$ |
> |  ----  | ----  | ----  | ----  |----  | ----  |----  | ----  |----  | ----  |----  | ----  |
> |  fully manual annotation  | 158k  |  1137k  |ImageNet |54.4  |20.5 |72.4  |90.9  |62.8  |35.5  |80.8 |94.0  |
> |  fully manual annotation  | 158k  |  1137k  |scratch |52.5  |23.5 |67.6 |86.3  |62.3  |34.9  |80.0 |93.3  |
> | 10% manual annotation  | 15k  |  127k  |ImageNet |33.8  |5.5 |45.3  |74.9  |36.1  |9.7  |48.6 |76.7  |
> | 10% manual annotation  | 15k  |  127k  |scratch |31.6 |5.7 |42.5 |72.2  |35.9  |8.6  |47.7 |75.3  |
> | ClusterNet (w/ gt sceneflow)  | 158k  |  0  |scratch |51.8 |21.3 |70.2 |89.5  |60.8  |30.2  |81.2 |94.8  |
> | ClusterNet  | 158k  |  0  |scratch |43.2 |18.4 |56.5|81.8 |55.4  |26.7  |71.9 |93.1  |
>
> Weakness: Since unsupervised 2D object detection without any pre-training is a challenging task, only a few previous works have explored this field. In Appendix B Table 1, we tried to compare with more methods to verify the effectiveness of our method. The significant gain comes from combining 2D and 3D information (also temporal information). Our unsupervised method showed a significant improvement compared with those 2D methods. Recently, the authors of FreeSolo (CVPR2022) released their code, and we added the experiment results of their method for comparison. Results have already been updated in Table 1 in the revised manuscript. FreeSolo could only achieve a 1.0 AP score when using their approach to generate pseudo masks and retraining the Faster R-CNN. The reason for the low AP score is that such attention-based methods are just semantic-level discrimination. A row of cars in the scene often generates a large mask, which can not distinguish specific instances. Therefore, they can perform well in datasets with a primary object, but their method can not be to directly applied in complex driving scenes.
>
> |  annotation setting   | #images  | #bboxes | network weights initialized from | AP$^{50}$ | AP$^{50}_{S}$ | AP$^{50}_{M}$ | AP$^{50}_{L}$ | AR$^{50}$ | AR$^{50}_{S}$ | AR$^{50}_{M}$ | AR$^{50}_{L}$ |
> |  ----  | ----  | ----  | ----  |----  | ----  |----  | ----  |----  | ----  |----  | ----  |
> | FreeSolo  | 158k  |  0  |ImageNet |1.0 |0.2 |1.0|1.9 |2.2  |0.0  |0.1 |12.7  |

---

> > ### Comment · Reviewer_nTyt · 2022-08-07
> > **Thank you for the response!**
> >
> > Hi,
> >
> > The provided response answer my questions. Thanks!

---

### Meta-Review · Area_Chair_2y8X · 2022-08-26

**Recommendation:** Accept
**Confidence:** Certain

**Metareview:**

This paper focuses on expanding the problem of unsupervised object discovery (detection) to a new setup, where a 3D point cloud is available as well as an RGB sequence. The paper received three detailed reviews from expert reviewers, all of which had their major concerns about the paper resolved through the author rebuttal and author-reviewer discussion period. With the extra analyses and experiments presented in the discussion period, the paper has reached the level of impact and contribution expected by NeurIPS papers. The authors are recommended to add these extra items to the final version of the paper.

**Award:**

No

---

### Decision · Program_Chairs · 2022-09-14

Accept